



# Seasonal and interannual variability of freshwater sources for Greenland's fjords

Anneke L. Vries[1,2], Willem Jan van de Berg[2], Brice Noël[3], Lorenz Meire[1,4], and Michiel R. van den Broeke[2]

[1]Department of Estuarine and Delta Systems, NIOZ Royal Netherlands Institute of Sea Research, Yerseke, The Netherlands
[2]Institute for Marine and Atmospheric Research, Utrecht University, Utrecht, the Netherlands
[3]Department of Climatology and Topoclimatology, University of Liège, Liège, Belgium
[4]Greenland Climate Research Centre (GCRC), Greenland Institute of Natural Resources, Nuuk, Greenland

**Correspondence:** Anneke L. Vries (anneke.vries@nioz.nl)

**Abstract.** The magnitude, source, release location, and timing of freshwater fluxes that end up in the numerous Greenland fjords is of special interest for ice-ocean interactions and ecosystems. In this study, we investigate intra- and interannual variability of the various freshwater sources for Greenland's fjords in seven climatologically distinct regions. For this, we use direct and statistically downscaled output from regional climate models for the mass fluxes, process-based estimates of basal melt and observational data for solid ice discharge. For the period 1940/1958 to 2023, we separately quantify runoff from the Greenland ice sheet, peripheral ice caps and tundra regions, and precipitation directly falling in the fjords. From 2009 onwards, the available data allows us to resolve the full seasonal cycle of freshwater fluxes. The results indicate a diverse range of relative contributions from freshwater sources between seasons and regions. Freshwater input in fjords in the wet southeast and northwest is dominated by solid ice discharge (55 % and 67 %, respectively) with a small contribution of tundra runoff, whereas in the relatively drier north, northeast, and southwest the contribution of tundra runoff is more important (20 %, 25 % and 30 %, respectively). Precipitation in fjords and tundra runoff can represent a large fraction of the monthly total, i.e. up to 11 % and 35 %, respectively, for winter and spring. However, the relative contribution of tundra runoff has been decreasing in time, the result of rapid increases in ice sheet and ice cap runoff over the past decades following atmospheric and oceanic warming. We show that the regional glacier-integrated melt-over-accumulation ratio (MOA) is a good predictor for the relative contributions of solid ice discharge, tundra runoff, and ice sheet runoff. These findings have implications for the use of freshwater fluxes forcing in regional ocean models and fjord studies, and enhance our understanding of their impact on ocean and fjord circulation and biogeochemistry.

## 1   Introduction

Greenland fjords constitute the hydrological connection between the Greenland Ice Sheet and the surrounding ocean. Of special interest for ice-ocean interactions and ecosystems are the source, magnitude, release location and timing of freshwater fluxes that end up in the numerous Greenland fjords. The source of freshwater input influences both the timing and location of its entry into the fjords. For instance, tundra runoff typically enters the fjord near the surface and peaks in spring when the





seasonal snow pack melts, whereas in glacial fjords runoff can both enter fjords at the surface or subglacially, at or above the glacier grounding line, with a pronounced peak in summer (Sanchez et al., 2023). These differences in magnitude and timing

of freshwater input result in different fjord circulation regimes (Mortensen et al., 2014), altering primary productivity, $CO_2$ uptake, cloud formation and ecosystems in fjords with marine-terminating glaciers compared to those without (Meire et al., 2017; Stuart-Lee et al., 2021; Meire et al., 2023; Wieber et al., 2024). Depending on the origin of the freshwater source, the concentration of nutrients and organic carbon can vary substantially (Hopwood et al., 2020).

Recent and future increases in freshwater discharge from the contiguous Greenland ice sheet (GrIS) and smaller glaciers and
ice caps (GIC) (Van den Broeke et al., 2009, 2016; Mankoff et al., 2021; Khan et al., 2022) have the potential to significantly affect the regional and large-scale ocean circulation, however specific timing, geographic location, and release depth are often not considered in global climate models (Yang et al., 2016; Dukhovskoy et al., 2019; Martin and Biastoch, 2023). To date, no global models and only a few regional models have a sufficient spatial resolution to resolve Greenland's fjords (Hallberg, 2013; Hewitt et al., 2017; Fox-Kemper et al., 2019; Gelderloos et al., 2022). Most of these models assume that i) freshwater enters
the fjords and the ocean at near-surface levels, ii) freshwater input is constant throughout the year, or iii) freshwater storage in fjords is negligible (Jackson et al., 2017; Dukhovskoy et al., 2021; Jackson et al., 2023; Martin and Biastoch, 2023) . Often, these assumptions are inaccurate. It has been shown that in west Greenland fjords, freshwater storage delays its export to the continental shelf by one to several months, depending on the fjord geometry and season, and the presence of land- or marine-terminating glaciers (Gladish et al., 2015; Stuart-Lee et al., 2021; Sanchez et al., 2023). In glacial fjords, a positive feedback
exists between the strength of the fjord circulation and glacial melt (Zhao et al., 2022), which implies that the magnitude and timing of freshwater input co-determines the total freshwater export onto the continental shelf.

To improve understanding and support modelling efforts, freshwater fluxes into Greenland fjords have been quantified through observations and models for Greenland as a whole and locally for individual fjords. Although useful for budget studies, Greenland total freshwater flux provides a limited perspective, as freshwater that leaves the fjord is transported in boundary
currents along Greenland and might end up in a deep convection region far from where it was released (Gillard et al., 2016; Duyck and De Jong, 2023). The different freshwater sources have different mixing depths which further affects their pathways, e.g. one model study assumed freshwater to be distributed over the upper 77 m on the shelf, which in case of dominating solid ice discharge may be too shallow and for tundra and GIC runoff too deep (Gillard et al., 2016). Therefore, models aiming to accurately simulate freshwater pathways should consider the type of freshwater input. To this end, recent spatially resolved
studies combine different datasets of freshwater fluxes from marine-terminating glaciers (Slater et al., 2022; Karlsson et al., 2023). These studies excluded the magnitude of terrestrial (tundra) runoff, which is perhaps the least known freshwater source (Marson et al., 2021). Bamber et al. (2018) estimated that tundra runoff adds 14 % to the total Greenland freshwater budget (2010-2016), with an updated estimate that is 62 Gt $yr^{-1}$ higher than the previous tundra runoff estimate ($\sim$ 25 % of the total) (Igneczi and Bamber, 2024). Some studies attempt to estimate freshwater fluxes into a limited number of individual fjords,
studying freshwater retention, salt and heat budgets and the impact of increased runoff on primary production (Jackson and Straneo, 2016; Oksman et al., 2022; Sanchez et al., 2023).





This study aims to improve and interpret Greenland-wide and regional estimates of freshwater sources for Greenland's fjords. To assess their relative importance, we combine high spatial and temporal resolution model data of precipitation, GrIS and GIC surface runoff, and basal melt with observational solid ice discharge estimates. In addition, we quantify tundra runoff

originating from rain and seasonal snow melt. This enables us to assess on a regional scale how the distribution between freshwater sources changes throughout the year, and how and why the magnitude and (seasonal) freshwater input into Greenland fjords have fluctuated over the past decades.

After introducing the methods and data sources in Section 2, we discuss the yearly sums and seasonal cycles of different freshwater sources in Section 3 and finish with a comparison to various other datasets, and climatological interpretation in

Section 4, and a summary in Section 5.

## 2 Methods

### 2.1 Data sources

To estimate sources of freshwater (FW) input into Greenland fjords, we combine different models and observational products (Table 1). As a default for runoff, melt, sublimation and precipitation, we used output from the polar regional atmospheric

climate model RACMO2.3p2 (Noël et al., 2019), RACMO henceforth. The used run was forced at three-hourly intervals (six-hourly before 1990) by the fifth generation re-analysis (ERA5) of the European Centre for Medium-Range Weather Forecasts (ECMWF), and starts in 1958. Over Greenland, RACMO runs at a native spatial resolution of $\sim$ 5.5 x 5.5 $km^2$(Noël et al., 2019). The output of this simulation is further statistically downscaled to a 1 x 1 $km^2$ polar stereographic grid (Noël et al., 2016, 2019).

For comparison with RACMO precipitation, the Copernicus Arctic Regional Reanalysis (CARRA-West), CARRA henceforth, is used, which has a 2.5 x 2.5 $km^2$ resolution and uses the HARMONIE-AROME model, cycle Cy40h1 (Schyberg et al., 2020; Box et al., 2023). CARRA-West runoff is unrealistically small over land and ice (i.e. <10 % of the RACMO estimate) and is not used here.

For comparison with RACMO runoff, we use outputs from the Modèle Atmosphérique Régional (MAR) version 14 forced

at six-hourly intervals by ERA5 since 1940, MAR henceforth. MAR runs at 5 x 5 $km^2$ spatial resolution and is statistically downscaled to the same 1 x 1 $km^2$ grid as RACMO (Fettweis et al., 2020).

Basal melt on a 1 x 1 $km^2$ grid is taken from Karlsson et al. (2021), which used a composite of estimates for basal melt from geothermal heat, friction and heat from surface meltwater, with the latter contributing to seasonal variability. Monthly values represent 1991–2023 averages. Basal melt has a total uncertainty of 21 %.

The FW components presented here, and discussed in detail in the following subsections, are GrIS runoff, GIC runoff, tundra runoff, basal melt, GrIS solid ice discharge (which represents the flux of solid ice across the grounding line, i.e. combining calving and submarine melt), and direct precipitation spatially integrated over the fjord surfaces. These components are presented as yearly sums since 1940/1958 and monthly sums since 1990 and are aggregated regionally (see below for region definitions) unless specified otherwise. The differences between the default products (boldface in Table 1) and other products



are typically small. That is why the default values are being reported in the figures and main text, and the differences with other products are discussed in Sect. 4.5.

### 2.1.1 Runoff

RACMO (1958-2023) provides the default products for GrIS and GIC runoff, while MAR runoff outputs (1940-2023) are used for comparison. Both have been statistically downscaled to 1 x 1 $km^2$ for the given period. The uncertainty of previous downscaled RACMO runoff products was estimated to be 20 % for GrIS and 40 % for GIC (Noël et al., 2019). RACMO also provides the default product for tundra runoff, which from the native RACMO 5.5 x 5.5 $km^2$ resolution is reprojected to the 1x1 $km^2$ grid using the nearest neighbour method. Tundra runoff is not statistically downscaled and therefore available since 1940. The tundra runoff uncertainty is estimated to be 10 % (Bamber et al., 2018). We do not apply a routing delay, i.e., all runoff is assumed to enter the fjords immediately, with the exception of ponding that is allowed at the ice surface in MAR, based on Zuo and Oerlemans (1996). Lakes are not represented in these versions of RACMO and MAR.

### 2.1.2 Precipitation

Precipitation into fjords is taken from two data sources. The default is the native RACMO precipitation (rain + snow) product (1958-2023), linearly interpolated onto a 1 x 1 $km^2$ grid (Huai et al., 2022), and the second source used for comparison is CARRA. We assumed both snow and rain to directly contribute to the freshwater input to the fjords, i.e. we neglected the precipitation phase and the possible storage effect of mass accumulation on top of sea ice.

When using the CARRA data, the difference between the 30-hour and 6-hour accumulated precipitation fields resulted in the daily precipitation, to account for spin-up issues of the water cycle as recommended by the developers. The CARRA time span is 1991-2023, and we reproject the monthly and yearly sums on a 1x1 $km^2$ grid using nearest neighbour interpolation. The uncertainty for the precipitation products is not given, and therefore assumed to be represented by the difference between RACMO and CARRA (15 %).

Precipitation was masked to only include fjords. We selected fjords by creating a convex hull around Greenland, and Qeqertarsuaq (Disko Island) separately, using Python's scipy.convexhull following the procedure proposed by Slater et al. (2022). We chose values for the parameters concavity (2) and length threshold (10), and removed all small clusters of grid cells (< 30 $km^2$) that were not branches of fjords (defined by calculating the fraction of land in a 4 km radius and setting the threshold to 75 % to exclude small bays that would have almost 50 % ocean in its surroundings). Then we removed all "land" points using the RACMO Land Sea Mask on a 1x1 $km^2$ resolution, leaving us with "sea" points that we classified as fjords. For a sensitivity analysis of fjord extent in the Discussion (Sect. 4.4), the convex hull was extended from only the main island to also include small offshore islands.





### 2.1.3 Solid ice discharge

For solid ice discharge, the default dataset consists of monthly values since 1986 for individual marine-terminating glaciers and is grouped per region (Mankoff et al., 2019). Winter observations from before 2009 are scarce, and missing values were linearly interpolated for annual values. However, monthly means for the seasonal cycle of relative contributions are calculated from 2009 onwards. The dataset excludes calving from GIC which is considered small (see Sect. 4.6). A similar procedure is used for a second solid ice discharge dataset from King et al. (2020), which has larger seasonal variability and is used for
comparison in the discussion (Sect. 4.5).

### 2.2 Region definitions

Greenland is divided into seven climatologically distinct regions (Fig. 1): North (NO), North-East (NE), North-West (NW), Central-East (CE), Central-West (CW), South-West (SW) and South-East (SE), mostly based on the seven land/ice basins from Slater et al. (2020) that in turn are based on the Mouginot et al. (2019) ice divides. We made one adjustment by moving
the boundary between SE and CE northward to better follow hydrological catchments, as well as to make the regions more comparable in size. Place names used in this study follow the convention by Oqaasileriffik, which has a map available on their website (Oqaasileriffik (The Language Secretariat of Greenland), 2024)

### 2.3 Melt-over-accumulation ratio

For interpretation purposes (see Sect. 4.2) the melt-over-accumulation (MOA) ratio is calculated using the following equation
applied to the GrIS and GIC mask in the seven basins:

$MOA = (\text{melt} + \text{rain})/(\text{snowfall} - \text{sublimation})$

All parameters have units mmWE and are taken from RACMO downscaled to 1x1 $\text{km}^2$ on the ice-covered surfaces. Rain is defined as total precipitation minus snowfall.

## 3  Results

### 140  3.1  Greenland integrated freshwater fluxes into fjords

Figure 2 shows a time series of the different components of the total annual freshwater input into Greenland's fjords, using six different data sources (see methods). Combined, the annual total FW flux into Greenland fjords averages 1144 ($\pm$ 170) Gt $\text{yr}^{-1}$ (0.035 Sv) between 1990 and 2023, where $\pm$ in this section represents uncertainty described in the methods (Sect. 2). The main contributor is solid ice discharge (462$\pm$ 43 Gt $\text{yr}^{-1}$), accounting for 40 % of the total. On average, GrIS (meltwater)
runoff is the second-largest source of freshwater into the fjords (31 %, 357 $\pm$ 53 Gt $\text{yr}^{-1}$), but in years with extreme surface melt, such as 2010, 2012, and 2019, GrIS runoff exceeds the contribution of solid ice discharge. The third-largest contributor



**Table 1.** Data sources used in this study. Data sets in boldface are used as default data in the results (Sect. 3), while the others are used for sensitivity analysis in Sect. 4.5.

| *Source* | *Name* | *Citation* | *Native resolution* | *Period* |
|---|---|---|---|---|
| **Runoff ice sheet (GrIS) + ice caps (GIC)** | **RACMO2.3p2 → 1 km** | **Noël et al. (2019)** | **5.5x5.5** $km^2$ | **1958-2023** |
| **Runoff ice sheet (GrIS) + ice caps (GIC)** | MAR3v14 → 1 km | Fettweis et al. (2020) | 5x5 $km^2$ | 1940-2023 |
| **Tundra runoff** | **RACMO2.3p2** | **Noël et al. (2019)** | **5.5x5.5** $km^2$ | **1940-2023** |
| **(Fjord) precipitation** | **RACMO2.3p2 → 1 km** | **Huai et al. (2022)** | **5.5x5.5** $km^2$ | **1958-2023** |
| (Fjord) precipitation | CARRA West | Køltzow et al. (2022) | 2.5x2.5 $km^2$ | 1991-2023 |
| **Solid ice discharge** | **Solid ice discharge** | **Mankoff et al. (2019)** | **n.a.** | **1986-2023** |
| Solid ice discharge | Solid ice discharge | King et al. (2020) | n.a. | 1985-2018 |
| **Basal melt** | **Basal melt** | **Karlsson et al. (2021)** | **1x1** $km^2$ | **2000-2020** |

is tundra runoff (15 %, 175$\pm$ 17 Gt $yr^{-1}$), followed by GIC runoff (6 %, 71$\pm$ 28 Gt $yr^{-1}$), and (fjord) precipitation (5 %, 54$\pm$ 8 Gt $yr^{-1}$). Basal melt contributes 2 %, (23$\pm$ 5 Gt $yr^{-1}$).

The period with data availability for all default components spans from 1990 to 2023, allowing for the calculation of absolute

and relative changes in annual freshwater sources over time. Total average FW flux since 2010 is 1239 ($\pm$ 180) Gt $yr^{-1}$, calculated to compare to other studies in Sect. 4.1. The freshwater sources that are increasing most rapidly since 1990 are GrIS and GIC runoff, whose relative contribution to the total fjord freshwater input increases by 0.17 % per year (p=0.04) and 0.05 % per year (p=0.002), respectively. Tundra runoff is also increasing in absolute terms, but slower than the other components, resulting in a decrease in its relative contribution of 0.08 % per year (p=0.002). Precipitation in fjords is changing very slowly

in an absolute sense, i.e. its relative contribution is decreasing by 0.03 % per year (p=0.01).

For the period when monthly resolution is available for all components (2009-2023), Fig. 3 shows the average seasonal cycle of the Greenland total freshwater fluxes into fjords based on monthly sums using absolute linear (a), absolute logarithmic (b), and (c) relative scales. Figure 3c is based on the default datasets (Table 1) and values are presented in Tables A1 and A2. Most freshwater input occurs during July at the peak of the melt season (296$\pm$ 57 Gt) and least during March in late

winter (48$\pm$ 4 Gt). Freshwater input during winter months is primarily driven by solid ice discharge (82$\pm$ 0.4 %, 41$\pm$ 1 Gt $month^{-1}$, DJF), whereas summer freshwater input is dominated by GrIS runoff (up to 68$\pm$ 3.5 %, 172$\pm$ 28 Gt $month^{-1}$, in July). Greenland tundra runoff peaks in early summer, when the seasonal snow melts (Fig. 3a) with a maximum of 36$\pm$ 3 % of the total freshwater input occurring in May (Fig. 3c). While direct precipitation in fjords contributes little on average, when examined at a monthly scale, it can account for up to 11$\pm$ 2 % of the total in January (Fig. 3c). Basal melt accounts for a

maximum of 3$\pm$ 0.6 % in March.



## 3.2 Regional freshwater fluxes into fjords

In order to study how the sources of freshwater are influenced by different climatic conditions across Greenland, surface types, (sea) ice conditions, and fjord geometry, Greenland is divided into seven different regions (Fig. 1). First, we discuss the time series of annual totals per region (Fig. 4), then the seasonal cycle focusing on the relative contributions of the FW fluxes (Fig. 170 5). In the following paragraphs, we will report 1990-2023 annual means of datasets indicated in boldface in Table 1, with a standard deviation of the annual values ($\pm$), unless reported differently. The absolute and relative average values can be found in Table A3 and A4, respectively. These results will be put into a climatological context in the discussion (Section 4.2).

The time series of annual FW fluxes show large variations between the regions (Fig. 4). In NO (Fig. 4a), which in absolute sense provides the least FW to fjords of all basins, all FW sources except (fjord) precipitation increase over time, with a 175 significantly increasing trend of 1.1 ($\pm$ 0.2) Gt yr$^{-2}$ for the total (p<0.001) (1990-2023). This region typically has large interannual variability in all runoff sources. At the start of the period, solid ice discharge was dominant (24$\pm$ 2 Gt yr$^{-1}$, 1990-2004), but later in the period GrIS runoff took over as the dominant FW flux in NO fjords (36$\pm$ 11 Gt yr$^{-1}$, 2005-2023). In NO, tundra runoff has a relatively large share in the total regional freshwater flux (17$\pm$ 3 Gt yr$^{-1}$) compared to other regions, associated with the relatively large tundra area in this dry part of Greenland, where the ice sheet is mainly land-terminating (Fig. 180 1b). The pattern for NE (Fig. 4b) is similar to the pattern for NO, with the exception of a relatively smaller role for GIC runoff (12$\pm$ 4 Gt yr$^{-1}$), and even more tundra runoff (30$\pm$ 5 Gt yr$^{-1}$), making the latter occasionally the largest FW contribution in this region. In contrast, the wetter NW (Fig. 4c) has glaciers that are mainly marine-terminating, with a relatively narrow tundra and a small fjord area (Fig. 1b). Here, the FW fluxes into fjords are clearly dominated by solid ice discharge (100$\pm$ 8 Gt yr$^{-1}$), which is greater than the combined contributions of all other sources. GrIS runoff accounts for approximately 30 % of 185 the input, while the other sources are relatively small. The NW solid ice discharge has been increasing since the early 2000s, in line with earlier studies that diagnosed the cause to be increasing ocean temperatures (Wood et al., 2018).

In CW (Fig. 4d), the contribution of GrIS runoff (63$\pm$ 18 Gt yr$^{-1}$) has been increasing, but with a high interannual variability. It is now occasionally larger than solid ice discharge, which remains the overall main contributor (83$\pm$ 9 Gt yr$^{-1}$). In contrast to most other regions, solid ice discharge decreased between 2013 and 2018 (Fig. 4d). Tundra runoff is relatively large in this 190 region (21$\pm$ 3 Gt yr$^{-1}$). Basal melt, (fjord) precipitation and GIC runoff are of similar small magnitude, contributing each 3 % to the annual total. In CE (Fig. 4e), the largest input source is solid ice discharge (77$\pm$ 6 Gt yr$^{-1}$), closely followed by GrIS runoff. Tundra runoff is approximately twice as large as GIC runoff (11$\pm$ 2 Gt yr$^{-1}$) or (fjord) precipitation (14$\pm$ 3 Gt yr$^{-1}$). This is the region where (fjord) precipitation has the largest relative contribution (up to 20 %), partly due to the relatively large fjord area (Fig. 1c). The largest and increasing contributor in the SW (Fig. 4f) is GrIS runoff (80$\pm$ 23 Gt yr$^{-1}$). Especially 195 before 2006, tundra runoff (55$\pm$ 9 Gt yr$^{-1}$) incidentally exceeded GrIS runoff during cold summers. Finally, the wet SE (Fig. 4g) has a narrow ablation zone (O (10 km)) and the FW flux into SE fjords is dominated by solid ice discharge (138$\pm$ 7 Gt yr$^{-1}$). Similar to NW, solid ice discharge in SE has been gradually increasing since the early 2000s (Fig. 4g).



### 3.2.1 Seasonal cycle of relative freshwater contributions per region

To further characterize the FW sources geographically and temporally, Fig. 5 shows the seasonal cycle per region based on
the relative monthly contributions since 2009, for data sets indicated in Table 1. In all regions, GrIS runoff is the dominant
freshwater source in July and August (Fig. 5), underlining the non-uniform distribution of this FW flux through the year,
with a strong summer peak. In some regions (SE, NW) with high-accumulation and/or narrow ablation zones, however, the
contribution of GrIS runoff in summer is only slightly larger than that of solid ice discharge in the same months (Fig. 5b,g).
Both GrIS and GIC runoff have their largest contributions in July, but their relative contribution varies per region. In NO, the
latter accounts for up to 25 % of the annual freshwater flux (Fig. 5a), while in NW, it does not exceed 5 % (Fig. 5b). The
relative contribution of solid ice discharge is always important in winter when the runoff fluxes are small. Regional variations
occur during summer, when the solid ice discharge share is low in the SW (Fig. 5f), while in other regions (NW, SE) it remains
the second FW source throughout summer (Fig. 5b,g). The largest share of tundra runoff of all seasons and regions (69 %) is
in May in SW (Fig. 5f). In the SW, tundra runoff even contributes in winter leading to the highest annual average (31 %), due
to the relatively high proportion of non-glaciated areas in the region and the year-round occurrence of seasonal snow melt (Fig.
1b). In other regions, the tundra runoff contribution to total FW input also peaks in May. There are similarities between the
seasonal cycle for each region, and many sources peak in the same month. However, the magnitude of this source's contribution
to the total flux, and its relevance at any given time, varies.

### 3.2.2 Trends in seasonal freshwater fluxes

Using the monthly data from 1990 to 2023, (sub)seasonal trends in the various FW components are examined. All FW sources
originating from land ice, i.e. GrIS solid ice discharge and GrIS and GIC runoff, are increasing in all regions and have been dis-
cussed previously. However, the patterns for tundra runoff and (fjord) precipitation are more ambiguous and will be discussed
in further detail below.

A positive trend in tundra runoff is found for CE, NW, NE, and SE in different seasons. In CE, an increase of $0.15\pm$
$0.04$ Gt $yr^{-2}$ (p=0.002) is observed, with more than half of the absolute increase occuring in summer (JJA). In NW, tundra
runoff increased by 9 % or $0.05\pm 0.01$ Gt $yr^{-2}$ (p=0.002), of which 0.03 Gt $yr^{-2}$ in JJA, (p=0.02), due to increasing annual
precipitation. In NE, tundra runoff has been increasing by $0.20\pm 0.09$ Gt $yr^{-2}$ (p=0.043). In SE there is a positive summer
(JJA) trend of 0.05 ($\pm 0.03$) Gt $yr^{-2}$ (p=0.07, 2.5 % of total annual precipitation). No significant trend was found for tundra
runoff in other seasons or regions.

Fjord precipitation mostly shows no significant trends except in CE and SE. CE has an increase in the months Sep-Oct of 0.04
Gt $yr^{-2}$ ($\pm 0.02$, p=0.02). In SE, there is a spring (MAM) increase of $0.04\pm 0.02$ Gt $yr^{-2}$, p=0.01) (3 % of total). In summary,
there is an increase in summer tundra runoff in CE, NW and SE, and in CE and SE there is a small (fjord) precipitation increase
in spring and fall.



## 4 Discussion

### 4.1 Comparison with other research

This study uses regional climate model data and observations of solid ice discharge to estimate the FW flux from different sources entering Greenland's fjords. We find an average total annual Greenland FW flux of $1.14 \times 10^3$ Gt yr$^{-1}$ for 1990-2023 ($1.24 \times 10^3$ Gt yr$^{-1}$ since 2010). FW fluxes per sector are SE 251 Gt yr$^{-1}$, SW 178 Gt yr$^{-1}$, CE 179 Gt yr$^{-1}$, CW 183 Gt yr$^{-1}$, and for the northern regions NE 119 Gt yr$^{-1}$, NW 146 Gt yr$^{-1}$, and NO 88 Gt yr$^{-1}$. We estimate uncertainty to be 20%, based on lower estimates for Greenland-wide surface mass balance. The average annual rate is slightly smaller than the total Arctic FW flux found by Bamber et al. (2018) (1300 Gt yr$^{-1}$ since 2010), which included non-Greenland ice caps. GrIS runoff between 2010-2016 is higher in Bamber et al. (2018) and Igneczi and Bamber (2024) than estimated in this study. Before 2000, GrIS runoff was significantly higher in Bamber et al. (2018) than in this study (∼350 vs ∼250 Gt yr$^{-1}$).

We find higher tundra runoff than Bamber et al. (2018) (150 vs 80 Gt yr$^{-1}$). Igneczi and Bamber (2024) estimated a total tundra runoff based on MAR between 140-160 Gt yr$^{-1}$ (1950-2021), similar to values found in this study using RACMO output. Observational studies show that tundra, GrIS and GIC runoff estimates are very sensitive to the selected region and are easily underestimated in regional climate models, with differences of up to 30-50 % between point measurements and regional climate models (Van As et al., 2014, 2018; Mankoff et al., 2020; Abermann et al., 2021). Yet, there are very few observational studies of total Greenland runoff and even less on tundra runoff, making estimates of the latter the least uncertain factor in the total FW input in its surrounding fjords and seas in this study.

### 4.2 Differences between regions and their implications

To determine potential drivers behind these regional differences, FW components were correlated with the melt over-accumulation (MOA) ratio (Sect. 2.3), based on land ice integrated mass fluxes in each region (Fig. 6a,b). We find that the fraction of total FW by solid ice discharge is strongly and negatively correlated with MOA (Fig. 6a), while tundra runoff fraction shows a strong and positive correlation (Fig. 6b). A somewhat weaker correlation exists between the fraction of total FW of GrIS runoff and MOA (r=0.78, p=0.04, not shown). Furthermore, the regional fractions to total FW of both GrIS and tundra runoff decrease with increasing fraction of solid ice discharge (Figs. 6c, d). These results can be intuitively understood: in a region with a smaller MOA ratio, i.e. experiencing relatively low melt and/or high accumulation, the ice sheet extends further towards and into the ocean. This leads to relatively narrow tundra and ablation zones, and a higher solid ice discharge contribution at the expense of GrIS and tundra runoff. In contrast to MOA, no clear correlations between the FW flux fractions and temperature and snowfall are found. Higher temperatures increase melt, hence GrIS runoff, and allow for a higher exposed land fraction, while more solid precipitation has the opposite effect. Both southern regions are relatively mild, but due to their different precipitation regimes, their FW flux fractions are different. Similarly, the NW sector is drier than the SW sector, but because of the lower temperatures, the FW fluxes in the NW are even more solid ice discharge-dominated than the SE sector.



## 4.3 Contribution of tundra runoff and (fjord) precipitation

Seasonal contributions from tundra runoff and fjord precipitation to Greenland-wide FW fluxes can reach up to 35% and 11%, respectively, exceeding the relative contributions reported in previous work (Bamber et al., 2018). The south and east of Greenland have relatively high precipitation rates (Fettweis et al., 2020; Van Dalum et al., 2024), which leads to the regions CE, SE, SW, and NE having a relatively high contribution of fjord precipitation, especially from October until April, when runoff fluxes are small (Fig. 5). This study also identifies an increase in summertime fjord precipitation in CE and SE, but no significant trends in annual totals are found. Although summertime fjord precipitation has a relatively low contribution to the total FW input, its impact differs from runoff as all precipitation enters the fjord surface waters directly in spatially relatively homogeneously fashion, thereby increasing stratification. While the precipitation phase (liquid or solid) is unlikely to have a large impact on FW input, it has a large impact on the heat budget. For example, snowfall into fjords can have a cooling effect on the upper layers, decreasing stratification, but this is outside the scope of this study.

In CE, NE, NW and SE we find an increase in (summer) tundra runoff. Compared to glacial runoff, tundra runoff brings a different type of nutrients to fjords, and its entry point is less concentrated than runoff and solid ice discharge from marine-terminating glaciers. This ultimately affects ecosystems (Hopwood et al., 2020), e.g., by shifting spring blooms to different communities. Despite the increase in absolute tundra runoff, its relative importance for the fjord freshwater balance is decreasing in time as FW sources from land ice are increasing more rapidly. An obvious reason is the limited water storage in the tundra seasonal snow cover compared to land ice.

## 4.4 Sensitivity to fjord definition

The region covered by a fjord is hard to define, and some freshwater studies only considered land surface basins (Khan et al., 2022), or in- or excluded large bays such as Disko Bay and Kangertittivaq (formerly known as Scoresby Sund). To address the sensitivity of our results to how fjords are defined, we performed a fjord area sensitivity test. When being more lenient in the algorithm outlining fjords to include bays such as Disko Bay and by running the algorithm not only for the mainland but also for all barrier islands, the total fjord (and bay) area increases by up to 87 % (Fig. A1). The largest relative increase in area is found in the NW (and NE), +177 % (+160 %), while in CE, the fjord area only increases by 7 %. This results in up to 186 % (181 %) increases in (fjord) precipitation in NE (NW), and 15 % more in CE. In NE and NW, this increases the relative contribution of (fjord) precipitation from 5 % to 8 %, and from 2 % to 3 % between 1990-2023. In CE, we find an increase from 8 % to 9 %.

## 4.5 Choice of data sources

In the previous sections, we used RACMO-based freshwater fluxes and solid ice discharge from Mankoff et al. (2019). As discussed in this section, using alternative estimates does not considerably change the absolute and relative contribution of the freshwater flux sources. CARRA gives slightly lower (fjord) precipitation values than RACMO (on average 7 Gt yr$^{-1}$ less, or 15 %), but the temporal variability is comparable (Fig. 2). The lower precipitation results in a decrease in the maximum





(monthly) share of fjord precipitation from 12 to 10 %. Regionally, the difference is largest in the SW in DJF (+3 %), which is true for most regions, except for NO where the largest reduction of the total share is in May (-3 %) (Fig. A2).

GrIS runoff from MAR is higher than RACMO (+3 %, 1990-2023), especially since 2010 (+5 %) (Fig. A3). This is not the
case for GIC, which is 3 % lower in MAR on average (1990-2023). The impact on the relative share of GrIS runoff to FW is only +1 % (1990-2023). Regionally, the differences between MAR and RACMO are most noticeable in NW (+8 Gt yr$^{-1}$), SW (+11 Gt yr$^{-1}$) and CE (-9 Gt yr$^{-1}$). The difference leads to a higher share of GrIS runoff when using MAR in NO, NW, and SW (+2.3 %, +3.6 %, and +3.4 % respectively) and a lower share in CE (-3.4 %). For GIC runoff, the largest change is -3 Gt yr$^{-1}$ in MAR in NE, reducing the relative share of GIC runoff by 1.9 % in NE.

Solid ice discharge in Mankoff et al. (2019) is only 0.9 % smaller than in the study by King et al. (2020). For their common epoch of monthly values (2009-2018) (see Sect. 2.1.3), the datasets agree on annual totals (Fig. 2). The latter dataset shows larger seasonal variability, resulting in a small maximum mean increase of 1 Gt in July over the whole of Greenland (Fig. A5, and a small relative change compared to the other components of 0.4 % in September (Fig. A5). Regionally differences are larger, leading to relatively less solid ice discharge in NO, NW, CE, decreasing the percentage share of solid ice discharge by
1.5 ($\pm$ 0.3) % on average (Fig. A5). In the other regions (NE, CW, SW, SE) the relative share of solid ice discharge increases by 0.6 ($\pm$ 0.3) % on average. The changes are not uniform over the months but are still considered small. We conclude that uncertainty due to use of different data sources is less than the interannual variability during the period 2009-2023.

### 4.6 Neglected processes and fluxes

Apart from the above-discussed uncertainties, this study covers freshwater input but not the freshwater budget of fjords. It
moreover neglects i) the storage effect of sea ice, i.e. collecting snow and rain accumulation, ii) the advection of sea ice into and out of the fjords, and iii) sea ice preventing precipitation from directly entering the fjord waters. However, sea ice melt constitutes a minor contribution to the total freshwater budget, and most sea ice is landfast and thus will melt in the same fjord where it originally formed (Cottier et al., 2010). Another process not considered is that the transformation from solid ice to FW may happen outside the fjord if icebergs leave the fjord (Moon et al., 2018). Finally, our study neglects FW storage
effects in the fjord, as fjord circulation is outside the scope of this study. We excluded solid ice discharge from the GIC, as it is estimated to be only 2.3-3.2 Gt yr$^{-1}$ between 2000-2020 averaged per decade (Kochtitzky et al., 2023), and the data set does not specify discharge on a regionally and monthly scale. The datasets for solid ice discharge use fixed flux gates and therefore do not account for the retreat or advance of the glacier front, which is estimated to be 63$\pm$ 6 Gt seasonally for the total ice sheet (Mankoff et al., 2020; Greene et al., 2024). This process is less important for a study addressing FW input than one addressing
the freshwater budget of fjords, as we assume fixed outlines of the fjord with a fixed flux gate.

### 5 Summary and conclusions

We estimated freshwater (FW) fluxes into Greenland fjords based on regional climate models (runoff, precipitation, melt-over-accumulation ratio), process-based estimates (basal melt), and satellite products (solid ice discharge). We individually



quantified the contributions of solid ice discharge from the contiguous Greenland ice sheet (GrIS), GrIS runoff, runoff from
peripheral ice caps and glaciers (GIC) and tundra runoff, and precipitation falling directly into the fjords. We provide a sea-
sonally resolved analysis from 1990 onwards, from when estimates of all contributing fluxes are available at a monthly time
resolution. We estimated that averaged over Greenland between 1990-2023, the relative contributions to the freshwater input
are 40 % for solid ice discharge, 33 % and 7 % for GrIS and GIC runoff, respectively, 15 % for tundra runoff and 5 % for
fjord precipitation. Considerable regional and seasonal variations exist. In winter, direct precipitation into fjords can contribute
up to 11 % to the total freshwater input Greenland-wide, and in May the relative contribution of tundra runoff peaks at 35 %.
The SE and NW regions of Greenland have a solid ice discharge-dominated FW regime, while the freshwater sources in the
SW are dominated by tundra runoff, GrIS and GIC runoff. The regional glacial melt-over-accumulation ratio is shown to be a
good predictor of the regional partitioning of freshwater fluxes into fjords; a high melt-over-accumulation ratio relates to a low
relative contribution of solid ice discharge and higher relative contributions of tundra and GrIS runoff. The large variability in
time and space of the FW fluxes into Greenland fjords needs to be taken into account to understand their present and future
impact on ecosystems and ocean circulation.

*Code and data availability.*   The (downscaled) RACMO and MAR data sets presented in this paper were previously published in Noël et al.
(2019) and Fettweis et al. (2020), and are available upon request and without condition from bnoel@uliege.be and xfettweis@uliege.be. The
solid ice discharge data is available from Mankoff et al. (2019) and King et al. (2020). The CARRA data is available through Schyberg et al.
(2020). The code and data required for figures are available from https://github.com/AnnekeV/Varia-Fresh-Fjords, and
https://doi.org/10.5281/zenodo.14551168, respectively.

**Appendix A:  Supplementary material**



**Table A1.** Average relative percentage per freshwater sources for the whole of Greenland between 2009-2023, in %.

| month | Solid ice discharge | Precipitation | Tundra runoff | GIC runoff | GrIS runoff | Basal melt |
|-------|---------------------|---------------|---------------|------------|-------------|------------|
| 1 | 81.3 | 11.3 | 3.3 | 0.3 | 0.9 | 3 |
| 2 | 83.5 | 9.9 | 2.5 | 0.2 | 0.8 | 3.1 |
| 3 | 84.8 | 7.8 | 3.1 | 0.3 | 0.9 | 3.1 |
| 4 | 74.3 | 7.3 | 13.6 | 0.5 | 1.6 | 2.8 |
| 5 | 49.2 | 4.5 | 34.5 | 1.4 | 8.4 | 2 |
| 6 | 23.3 | 1.8 | 29.2 | 6.4 | 37.9 | 1.5 |
| 7 | 12.8 | 1 | 12.5 | 12.7 | 60 | 1 |
| 8 | 19.8 | 2.2 | 11.7 | 11.2 | 53.6 | 1.5 |
| 9 | 50.9 | 7.9 | 12.2 | 4.7 | 21.8 | 2.6 |
| 10 | 73.6 | 10.6 | 9.3 | 0.7 | 2.9 | 2.8 |
| 11 | 78.1 | 10.2 | 7.3 | 0.3 | 1.2 | 2.9 |
| 12 | 81.5 | 9.5 | 4.8 | 0.3 | 1 | 3 |
| **Total** | 39.8 | 4.5 | 14.5 | 6.7 | 32.7 | 1.9 |







**Figure 1.** a) Masks of different surface types at 1x1 km$^2$ resolution representing: the contiguous ice sheet (GrIS, orange), tundra (green), glaciers and ice caps (GIC, purple) and fjords (red). Thick black lines delineate the seven climatological regions used in this study. b) Relative contribution of different regions to total surface type area. c) Total area per surface type and region.



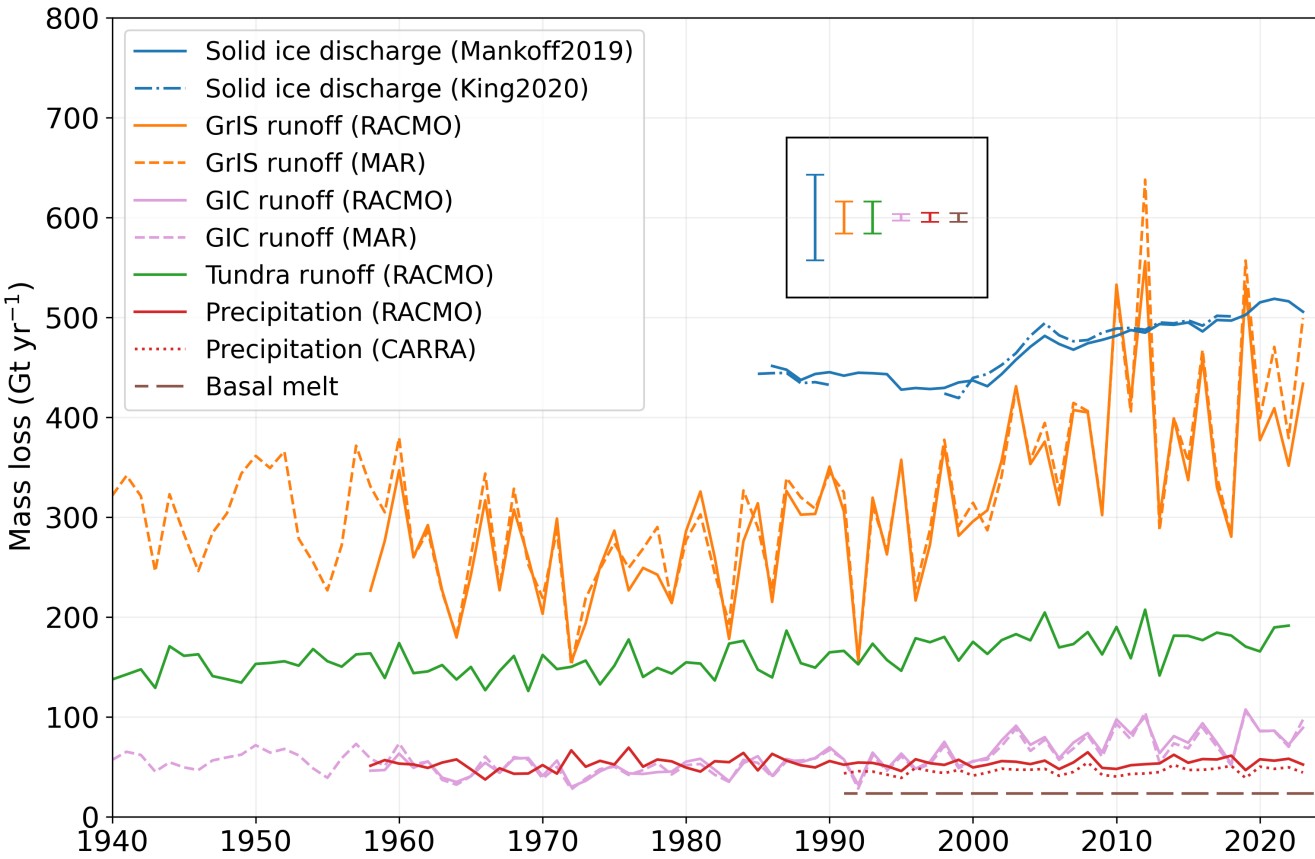

**Figure 2.** Time series of annual, Greenland total FW input into fjords for different components between 1940-2023. Error bars represent typical uncertainty per FW source.





**Figure 3.** Average seasonal cycle of components of Greenland total FW input with solid ice discharge (blue), GrIS runoff (orange), GIC runoff (purple), tundra runoff (green) and (fjord) precipitation (red) and basal melt (brown) a) Average seasonal cycle of FW components and total (black) for 1990-2004 (dashed) and 2005-2023 (solid). Shading represents standard deviation due to interannual variability. b) as in a) but with a log scale. c) Seasonal cycle of the relative source contributions to the total FW input between 2009-2023 (see Methods). Datasets used are indicated in bold in Table 1.





**Figure 4.** Time series of annual FW input per region between 1990-2023. Data sets used are indicated in bold in Table 1.





**Figure 5.** Seasonal cycle of the relative contribution of sources to the total FW input between 2009-2023 per region (see Methods). The datasets used are indicated in bold in Table 1.





**Figure 6.** Correlation plots for the seven regions between a) fraction (of total FW flux) of solid ice discharge and MOA, b) fraction of tundra runoff and MOA, c) fraction of solid ice discharge and fraction of GrIS runoff d) fraction of solid ice discharge and fraction of tundra runoff and their respective r and p values.





**Table A2.** Average absolute flux from FW sources into Greenland fjords between 1990-2023 with standard deviation of interannual variability ($\pm$), in Gt yr$^{-1}$.

| month | Solid ice discharge | Precipitation | Tundra runoff | GIC runoff | GrIS runoff | Basal melt | Sum |
|-------|---------------------|---------------|---------------|------------|-------------|------------|-----|
| 1     | 40.6$\pm$ 1.0       | 5.7$\pm$ 1.2  | 1.6$\pm$ 0.5  | 0.1$\pm$ 0.0 | 0.4$\pm$ 0.0   | 1.5$\pm$ 0.0 | 49.9$\pm$ 2.8  |
| 2     | 40.6$\pm$ 0.9       | 4.8$\pm$ 1.7  | 1.2$\pm$ 0.5  | 0.1$\pm$ 0.0 | 0.4$\pm$ 0.0   | 1.5$\pm$ 0.0 | 48.6$\pm$ 3.2  |
| 3     | 40.6$\pm$ 0.9       | 3.7$\pm$ 1.3  | 1.5$\pm$ 0.8  | 0.1$\pm$ 0.0 | 0.4$\pm$ 0.0   | 1.5$\pm$ 0.0 | 47.8$\pm$ 3.0  |
| 4     | 40.6$\pm$ 0.9       | 4.0$\pm$ 1.1  | 7.4$\pm$ 4.4  | 0.3$\pm$ 0.1 | 0.9$\pm$ 0.4   | 1.5$\pm$ 0.0 | 54.6$\pm$ 6.8  |
| 5     | 40.9$\pm$ 0.9       | 3.7$\pm$ 0.9  | 28.7$\pm$ 10.4 | 1.1$\pm$ 0.7 | 7.0$\pm$ 6.5  | 1.7$\pm$ 0.0 | 83.1$\pm$ 19.4 |
| 6     | 41.6$\pm$ 1.0       | 3.2$\pm$ 1.3  | 52.0$\pm$ 9.5 | 11.4$\pm$ 4.4 | 67.6$\pm$ 29.2 | 2.6$\pm$ 0.0 | 178.4$\pm$ 45.3 |
| 7     | 41.6$\pm$ 1.1       | 3.2$\pm$ 1.0  | 40.5$\pm$ 5.1 | 41.0$\pm$ 8.0 | 194.6$\pm$ 40.5 | 3.3$\pm$ 0.0 | 324.2$\pm$ 55.8 |
| 8     | 40.6$\pm$ 0.9       | 4.4$\pm$ 1.1  | 23.9$\pm$ 3.8 | 23.0$\pm$ 3.1 | 109.8$\pm$ 22.6 | 3.1$\pm$ 0.0 | 204.9$\pm$ 31.4 |
| 9     | 40.1$\pm$ 0.8       | 6.2$\pm$ 1.6  | 9.6$\pm$ 4.2  | 3.7$\pm$ 1.9 | 17.2$\pm$ 12.6 | 2.1$\pm$ 0.0 | 78.8$\pm$ 21.1 |
| 10    | 40.3$\pm$ 0.8       | 5.8$\pm$ 1.7  | 5.1$\pm$ 1.8  | 0.4$\pm$ 0.2 | 1.6$\pm$ 1.0   | 1.5$\pm$ 0.0 | 54.8$\pm$ 5.6  |
| 11    | 40.5$\pm$ 0.9       | 5.3$\pm$ 1.1  | 3.8$\pm$ 1.4  | 0.2$\pm$ 0.0 | 0.6$\pm$ 0.2   | 1.5$\pm$ 0.0 | 51.9$\pm$ 3.5  |
| 12    | 40.6$\pm$ 0.9       | 4.7$\pm$ 1.1  | 2.4$\pm$ 0.7  | 0.1$\pm$ 0.0 | 0.5$\pm$ 0.0   | 1.5$\pm$ 0.0 | 49.9$\pm$ 2.8  |
| **Mean** | 40.7$\pm$ 0.9    | 4.6$\pm$ 1.3  | 14.8$\pm$ 3.6 | 6.8$\pm$ 1.5 | 33.4$\pm$ 9.4  | 1.9$\pm$ 0.0 | 102.2$\pm$ 16.7 |





**Table A3.** Average absolute fluxes from FW sources into fjords per region between 1990-2023, in Gt $\mathrm{yr}^{-1}$.

| month | Solid ice discharge | Precipitation | Tundra runoff | GIC runoff | GrIS runoff | Basal melt | Sum |
|---|---|---|---|---|---|---|---|
| NO | 25 | 2 | 17 | 12 | 30 | 2 | 88 |
| NW | 98 | 2 | 5 | 2 | 35 | 4 | 146 |
| NE | 25 | 6 | 30 | 13 | 43 | 3 | 119 |
| CW | 82 | 6 | 21 | 5 | 63 | 5 | 183 |
| CE | 74 | 14 | 22 | 11 | 57 | 3 | 179 |
| SW | 19 | 10 | 55 | 10 | 81 | 3 | 178 |
| SE | 139 | 14 | 26 | 19 | 49 | 4 | 251 |
| Total | 462 | 54 | 175 | 72 | 357 | 23 | 1144 |





**Table A4.** Average relative fluxes from FW sources into fjords per region between 1990-2023, in %.

| Basins | Solid ice discharge | Precipitation | Tundra Runoff | GIC Runoff | GrIS Runoff | Basal melt |
|:------:|:--------------------|:--------------|:--------------|:-----------|:------------|:-----------|
| NO | 28 | 3 | 20 | 14 | 34 | 2 |
| NW | 67 | 2 | 4 | 2 | 24 | 3 |
| NE | 21 | 5 | 25 | 11 | 36 | 3 |
| CW | 45 | 3 | 11 | 3 | 35 | 3 |
| CE | 41 | 8 | 12 | 6 | 32 | 1 |
| SW | 11 | 6 | 31 | 6 | 46 | 2 |
| SE | 55 | 6 | 10 | 8 | 19 | 2 |
| Total | 40 | 5 | 15 | 6 | 31 | 2 |



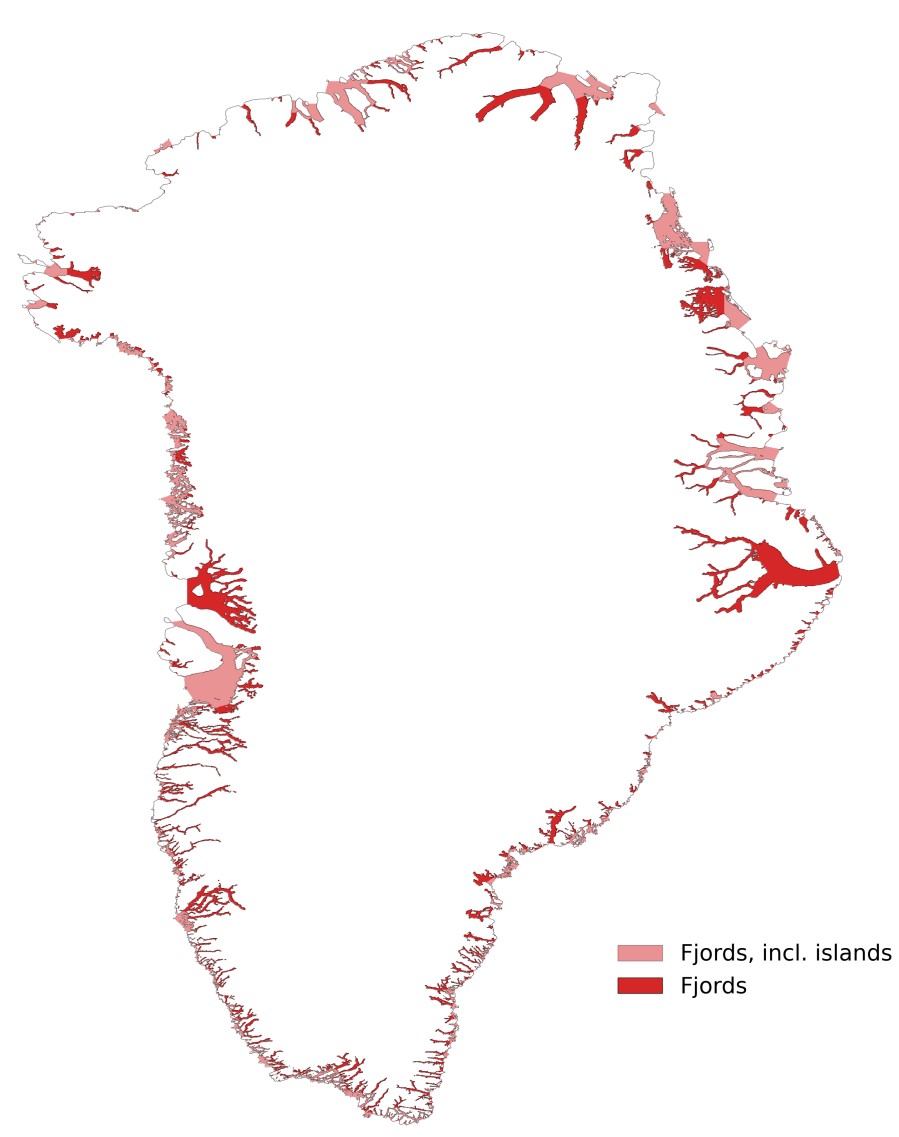

**Figure A1.** Fjord mask used in this study (dark red) and in light red large 'fjord' mask used for sensitivity, including islands and small bays. The fjord area is $75 \times 10^3$ km$^2$ in the default case and $140 \times 10^3$ km$^2$ when also large bays are taken into account.



**Figure A2.** Seasonal cycle of the relative contribution of sources to the total FW input between 2009-2023 per region, with (fjord) precipitation from CARRA-West.



**Figure A3.** Time series of annual freshwater input per region between 1990-2023, with GIC and GrIS runoff from MAR.





**Figure A4.** Average seasonal cycle of Greenland total FW input based on monthly averages with solid ice discharge (blue) by King et al. (2020), GrIS runoff (orange), GIC runoff (purple), tundra runoff (green) and fjord precipitation (red) and basal melt (brown) a) Average seasonal cycle for 1990-2004 (dashed) and 2005-2023 (solid). Shading represents standard deviation due to interannual variability. b) as in a but with a log scale. c) Seasonal cycle of the relative source contributions to the total FW input between 2009-2023.




**Figure A5.** Seasonal cycle of the relative contribution of sources to the total FW input between 2009-2023 per region, but with solid ice discharge by King et al. (2020).





*Author contributions.* AV, WJB and MB designed the study and AV carried it out. BN provided regional climate model data. All co-authors were involved in the scientific discussions that led to this study. AV prepared the manuscript with contributions from all co-authors.

*Competing interests.* At least one of the (co-)authors is a member of the editorial board of The Cryosphere.

*Acknowledgements.* I acknowledge the use of LanguageTool for suggesting corrections for spelling and grammar mistakes in the final stage of preparing this manuscript.



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
