# Peer review of "Seasonal and interannual variability of freshwater sources for Greenland's fjords"

_EGUsphere, 2024_

## Author Comment (AC1)

**Reviewer 1**

First of all, we would like to thank Ken Mankoff for his time and fast response. We appreciate the feedback received, it has greatly improved the manuscript. Responses to the issues raised (in blue) are written in black, and changes in the manuscript are written in red.

L129-131 states "We made one adjustment by moving the boundary between SE and CE northward to better follow hydrological catchments, as well as to make the regions more comparable in size."
Must you do this? Please don't do this. I assume you are not trying to make your product less useful and usable and add confusion and extra effort for others, but without good justification for adding yet another new and different basin and region product, I cannot let this comment go. See https://x.com/glasheologist/status/1557208390853505024 There are entire community projects (e.g., https://iacs-cryo.github.io/Delineation-WG/deliverable1) and many papers showing that comparing products A and B is hard or impossible or incorrect because they used different boundaries.

We understand your concern regarding the many different basin definitions that are being used in the community. The main reason for doing so is that, for the purpose of this paper, we are 'in-between' applications for land, ocean and ice. This study has the purpose of comparing freshwater sources from the perspective of fjords and climate, as opposed to hydrological basins (on land), ocean basins or glaciological basins. During our research phase, we concluded that using any of the existing basin definitions would not fit our research needs, and therefore a dedicated basin definition was deemed necessary, despite the disadvantage of creating another basin definition as you point out. We followed, as much as possible, existing definitions. We use as a basis the ocean-based basin divides used by Slater et al. (2020), which in turn represents an adjustment of the ice basin divides used by Mouginot et al. (2019). However, the area of CE is very small in Slater et al. (2020), as it was created as a 'transition' sector between colder and warmer ocean waters, while not cutting the Kangerlussuaq glacier in half (Fig. 1). This makes NE six times larger than CE, making the regions not comparable anymore. Furthermore, the Mouginot et al. (2019) - CE/SE boundary over the tundra cuts fjords in half, which is unsuitable for our purposes. To compensate for both disadvantages, we chose to follow the 'ice divide' boundary from Mouginot et al. (2019) to differentiate between CE/SE. Over the tundra, we manually adjusted the CE/SE boundary to follow the hydrological basin boundaries over land (see Figure 1 in this document, and Zenodo database, see comment "In addition to using standard regions,"). Figure 6 in the MS confirms that our updated basins have the desired inter-distinguishable hydrological and climatological characteristics, indicating that the chosen regions effectively capture variance across Greenland. We will provide information (see below) to increase reproducibility, and to make the outcomes comparable to other studies. We revised the manuscript as follows:

Because this study considers freshwater fluxes into fjords, which differs from most hydrological, glaciological and oceanic applications, Greenland is divided into seven climatologically distinct regions (Fig 1): North (NO), North-East (NE), North-West (NW), Central-East (CE), Central-West (CW), South-West (SW) and South-East (SE). These are based on the seven land/ice basins from Slater et al. (2020) that in turn are based on the Mouginot et al. (2019) ice divides. We made one adjustment by moving the boundary between SE and CE northward, following the Mouginot et al. (2019) ice divide, to follow tundra hydrological catchments, as well as to make the regions more comparable in size.

In addition to using standard regions, I would strongly encourage you to provide data (even if not results in the paper) at fjord scale. Or in map (spatial) form at GIS scale. Your work is only possible because of all of the data products you were able to ingest and process. You should strive to make your products even more usable than what you ingested - raise the bar if possible. While some of those products were specifically data products (and published in ESSD), many others were science-focused papers such as this manuscript but that also shared usable data. Data shared at fjord scale can be easily combined up to region or GIS scale. Data shared spatially at GIS scale (at a reasonably high resolution, e.g < 25 km grid cell) can be processed down to region or fjord resolution. The only way to have data less useful than providing it at regional scale is to provide a single value at GIS scale.
I know Mankoff products are available at fjord scale, Karlsson is GIS wide spatial format (can be accumulated at fjord scale), and as can your RCM inputs. I'm not sure if any of your inputs are provided at only regional scale.
In summary, use standard regions, and provide data everywhere in spatial format, or at glacier scale in vector format.
Thank you for your suggestion. We have not made individual fjord delineations for this study, only a fjord mask and basin delineations. We will provide the fjord mask and used basin definitions and freshwater data on 1 km scale as you suggested. This will allow others to use their own basin delineation and improve reproducibility. We will include that CARRA data are available from https://cds.climate.copernicus.eu/datasets/reanalysis-carra-single-levels.
The (downscaled) RACMO and MAR data presented in this paper were previously published in Noël et al. (2019) and Fettweis et al. (2020). The 1 km RACMO and MAR data used in this manuscript, as well as the region and fjord masks, are available from https://doi.org/10.5281/zenodo.14551168, respectively.

** MoA
Sect. 2.3 Annoying to see a new term and have to jump around to learn about it. Add a brief intro to MoA here? Anyway, 4.2 just refers straight back to 2.3. I still don't know what MoA is beyond a mathematical definition. You're introducing a new term and concept here (I think? I've never seen it and you have no citation for it). Add some text? What's the point of MoA? What do we learn from it? What are its weaknesses and limitations? What's a reasonable range for it? Min, max, average, median? How does it change spatially or temporally and why? Some of these are addressed in the text, but very few.
I've now spent some time thinking about MoA and trying to understand it, looking at Fig. 6, etc.

MoA = (melt + rain) / (snowfall − sublimation)

low MoA = low melt or high snowfall

high MoA = high melt or low snowfall

This seems to mostly be an atmospheric phenomenon? But perhaps controlled by elevation too? What about providing a spatial map of average MoA. Does something interesting pop out? I assume nothing interesting in the interior where MoA goes to 0 because of no melt. I'd expect regions with high winter snowfall to also have high summer melt (e.g., Southern Greenland) and regions with low accumulation to also have low melt (N. Greenland). The ablation area is where it gets interesting. Is it just a proxy for width of ablation zone? Something else?

I also assume it's only useful as an annual average, not monthly. How it changes in time is also unlikely to be interesting, as we know there's an increase in melt that is larger than the increases in snowfall. But maybe I'm missing something.

I think you need to introduce MoA as a stand-alone product before you start correlating it with external things like Discharge (Fig. 6a) at the beginning of Sect. 4.2.
We acknowledge that the introduction of MoA was somewhat ad hoc, and we regret introducing a term that is not well-known without providing appropriate explanation. We appreciate the reviewer drawing our attention to this issue, which we have resolved by revising the text as follows:
The MoA ratio has been used previously in firn studies to determine the climatic conditions under which melt would generate runoff in the accumulation zone. Previous work has identified a theoretical MoA ratio threshold between 0.6 and 0.7, indicating the onset of runoff (Pfeffer et al., 1991; Braithwaite et al., 1994). More recently, MoA has been used to predict when melt ponding starts on Antarctic ice shelves ( van Wessem et al., 2023). In this study, MoA is used as a climatological indicator over the ice sheet, which we hypothesise is highly relevant for the partitioning between solid ice discharge and liquid water runoff into fjords. MoA does not directly depend on runoff, but on melt as well as snowfall; the relative regional sizes of ablation and accumulation areas and the potential for meltwater buffering (through snowfall) also become important.

L255: To our knowledge, no studies have identified a strong link between MoA and freshwater input fractions in Greenland fjords. In contrast, freshwater input fractions into fjords poorly correlate with temperature, melt or snowfall. This novel result will facilitate the interpretation of e.g. future changes in the distribution of freshwater fluxes in terms of climate change.

What is the point of 6a? I think it's a complicated way of saying that when there is high melt, the proportion of discharge goes down.
See also answer RC2: L251-52 / Figure 6(c-d):
"We clarify in the revised text as follows: Furthermore, the regional fractions to total freshwater

input of both GrIS and tundra runoff decrease with increasing fraction of solid ice discharge (Figs. 6c,d). Yet, no such relation is found for fraction of GIC runoff or precipitation. There are more sources than solid ice discharge and meltwater runoff alone, such as ice cap runoff, tundra runoff and precipitation. This means that the relationship between two sources is not a priori linear, and we find that there is such a relation between all pairs of source fractions."

**\* Minor comments**

L24: How does something enter a fjord above a grounding line?
Thanks for pointing out this confusing sentence. We will change the sentence to:
(…) whereas in glacial fjords, runoff can enter at the surface and/or subglacially,

Paragraph starting L42: Seems like the text here should make reference to Mankoff (2020) http://doi.org/10.5194/essd-12-2811-2020 which uses your RCM (and MAR) to distribute both ice sheet and tundra runoff at stream resolution. L51 in particular, tundra runoff is not excluded by Mankoff (2020) http://doi.org/10.5194/essd-12-2811-2020. That paper does not "combine different datasets" (L50), but is trivial to combine with other Mankoff products to get solid discharge RCM runoff terms. I also note that the Mankoff discharge product provides an estimate of discharge depth for every stream outlet, addressing the surface vs subsurface discharge issues raised here.
We agree with this observation, and we have changed the reference as follows:
To this end, recent spatially resolved studies estimate freshwater fluxes from marine-terminating glaciers or stream outlets (Mankoff et al. 2020a; Slater et al., 2022; Karlsson et al., 2023).

L143: There is no explicit uncertainty section in Sect. 2, but it sounds like there should be? Is it 1 or 2 sigma?
In this case the uncertainty is the uncertainty as described per data source in section 2. We have changed the wording below:
(...), where the errors in this section represent the uncertainty per source as described in the methods.

L164: Direct inputs in January? Seems unlikely w/ ice cover? I realize you address this at the end of the document, but I think it should be addressed more explicitly throughout.
See answer on comment Sect. 4.6, and we added the following sentence to the method section:
Precipitation is defined as precipitation onto the fjord area, neglecting the potential temporary interference of sea ice that may be present, as discussed in Sect. 4.6.

L164: Basal amount is mostly steady state. What is the goal of reporting this small amount on this month? What is the significance of this sentence?
It is true basal melt is small with low temporal variability. The sentence is included because all terms are discussed, and we want to be consistent by discussing every term.

L240: Mankoff 2020 (freshwater) shows RCMs and obs disagree by >> 100%, not 30 - 50 %.
We changed the sentence as follows:
(...), with differences of up to 30-50 % between point measurements and regional climate models for larger regions, or even more than 100 % for smaller catchments (Mankoff et al., 2020).

Table A1 and elsewhere: Do you know things to 0.1 %?
Thank you for your suggestion. We removed decimals in Table A1 and elsewhere where relative numbers are reported, except for values smaller than 1, where we will keep one decimal.

What version of Mankoff (2019) data did you use? Maybe change Mankoff (2019) to Mankoff (2020) http://doi.org/10.5194/essd-12-1367-2020
We used version v100, from November 2024. We changed the citation to:
Mankoff, K. D., Solgaard, A., Colgan, W., Ahlstrøm, A. P., Abbas Khan, S., and Fausto, R. S.: Greenland Ice Sheet solid ice discharge from 1986 through March 2020 [v100], Earth System Science Data, 12, https://doi.org/10.5194/essd-12-1367-2020, 2020b

Your Zenodo dataset contains folders and files like ".DS_Store" and "__MACOSX/data/temp/._MoA_plot_input_mean_per_region_1990-2023.csv". Consider adding the DOI/URL of "all versions" that Zenodo provides and in the text mentioning which version you used. This way if you update the data in the future, readers can find the version in the paper or the latest version.
Thanks for pointing this out, and apologies. The dataset is more 'clean' now and the files created by Mac OS in the zipping process are removed, as well as introducing a version for every new 'update' of Zenodo.

"Flux": This is a technical term with specific units of mass or length^3 (volume) per unit time per unit area. You are never using per unit area, and should probably never use the word flux (except perhaps when referring to ice discharge flux gates). I believe the correct term is either "mass flow rate" if Gt/yr or "volume flow rate" if km^3/yr.
Thanks for pointing this out. In this study, we use the word "flux" to refer to freshwater entering fjords as mass per time unit. While we'll remove "flux" whenever feasible, the term has been used in recent literature to describe mass flow rate or volume flow rate (e.g. Bamber et al., 2018; Karlsson et al., 2023). In some cases, "flux" is more appropriate than alternative terminology, which is why we've included this specific definition at the beginning of our methods section:
Although the formal definition of flux is a volume per unit time per unit area, in this study, the term freshwater flux is used to indicate freshwater entering fjords as mass per time unit, similar to Bamber et al., (2018) and Karlsson et al., (2023).

What is sublimation? Is it the net term (deposition + condensation - evaporation - true sublimation), or is it true sublimation?
Sublimation is surface sublimation + sublimation of drifting snow - deposition. RACMO does not model phase changes between liquid and gas over glaciated surfaces like the ice sheet, so condensation and evaporation are zero.

Sect 4.6 It's great that this product addresses freshwater input to fjord surfaces. But I encourage you to be more up-front about the limitations of this new approach, the primary limitation being sea ice. It's the last section before conclusions, but is more important than that. Going back to highly precise uncertainty (fractions of a percent!), what exactly is your uncertainty measuring? Some arbitrary mathematical function of the RCM, or is it telling us something useful about what we do and do not know about freshwater fluxes at regional and temporal scale? If the former, that's easy but not terribly useful. If the latter, that may be more useful for downstream users, but your uncertainty in winter months must then increase because of the neglected sea ice processes. That is, perhaps you can address sea ice in this work in some manner, without adding a full model of sea ice growth, winter RCM accumulation onto the sea ice, and then sea ice advect & decay which allows the accumulated mass to then enter the fjord. That's out of scope for this paper, but I hope some treatment may not be out of scope.
We agree that the effect of sea ice deserves more up-front treatment in the manuscript. We extend Sect. 4.6 as follows:
The effect of sea ice on the fjord's freshwater budget varies per region and season. Most sea ice in Arctic fjords is landfast and thus will melt in the same fjord where it originally formed, temporarily storing freshwater rather than being a separate source (Cottier et al., 2010). Large regional and temporal variability exists in sea ice presence, and timing of formation and melt. In SE Greenland, Arctic sea ice can arrive from offshore, importing freshwater, while fjords in the SW can remain largely ice-free throughout the year (Moon et al., 2024; Stuart-Lee et al., 2021). The timing of seasonal sea ice break-up differs regionally: in SE Greenland, it occurs between May and August, while NO fjords can be ice-covered throughout most of the year (Moon et al., 2024; Johnson et al., 2011; Zahn et al., 2024).

We hypothesize that the uncertainty resulting from neglected sea ice processes will be larger with increasing latitude, more in the east than in the west and in winter months. Exploring the impact of sea ice on the freshwater budget is a potential avenue for future research.

Table 1: MAR "source" is bold but nothing else on that line is bold.
We have changed this accordingly.

Fig 2: y-axis units might be "Mass flow rate" or "volume flow rate" not "loss"?
See comment "Flux"; we changed the y-axis unit to Freshwater flux [Gt yr-1].

Fig 3: y-axis units are not Flux but Mass flow rate.
See comment "Fig. 2".

Fig 6a: Add units to y-axes. Replace symbols with letters (i.e "NO, "NW" in the plot, no legend needed).

Regarding the y-axes units, we'd like to clarify that the values represent fractions, which are dimensionless quantities. This information is already provided in the figure caption, but we'll ensure it's more clearly stated to avoid confusion.

Table A3, A4: If MoA is interesting and you keep it, should it be a column in these tables (region average MoA)?

Thanks for the suggestion. We think MoA is mostly interesting in predicting/explaining the other variables and is therefore less interesting as a standalone variable. For this reason, we think that the numbers that can be read from the figures are sufficient for the interpretation.

Competing interests: This is weirdly phrased. Why not state who has which specific competing interests?

Agreed, we changed it to:

BN and MB are members of the editorial board of The Cryosphere.

Ack: Cite LanguageTool (URL if no scientific paper)

Added

Cite software used.

Added to code/data availability

Data analysis and figure plotting was done using Python 3.12 and the map was made using QGIS (Van Rossum and L. Drake, 2009; QGIS Development Team, 2025).

Cite all data products DOIs, not just scientific papers. Mention versions of data products.

Changed this.

When you say "From 1990 to 2023" I don't know if this includes 2023 or not. I recommend "A through B" if you went to the end of B.

Changed this throughout the document.

[Figure]

*Figure 1: Masks of different surface types at 1x1 km2 resolution representing: the contiguous ice sheet (GrIS, orange), tundra (green), glaciers and ice caps (GIC, violet) and fjords (red). Solid black lines delineate the seven climatological regions used in this study, dashed olive green lines delineate the Slater et al. (2020) basins and dashed cyan blue lines delineate the Mouginot et al. (2019) basins (see legend in the upper left corner).*

**References:**

[revised manuscript text omitted]

---

## Author Comment (AC2)

**Reviewer 2**

First of all, we would like to thank the reviewer for their time. We appreciate the feedback received which has greatly improved the manuscript. Responses to the issues raised (in blue) are written in black, and changes in the manuscript are written in red.

L52-54: This sentence is quite confusing. I suggest instead writing what the tundra contribution is for the whole model time period for all these studies (Bamber et al., 2018, Mankoff et al., 2020, Igneczi and Bamber, 2024). Something along the lines of:
"Bamber et al., 2018, estimated that tundra runoff added on average ?? % from 1958-2016, while Mankoff et al., 2020 found a larger number of.. Igneczi and Bamber (2024) estimated an even higher contribution of [..]."
Thanks for the suggestion. We have removed the relative contribution, because it was not specifically mentioned in these studies.
We changed the text in the revised MS to:
Bamber et al. (2018) estimated total Greenland tundra runoff at approximately 80 Gt yr-1 (1958-2016), while Igneczi and Bamber (2024) found a higher number of 140-160 Gt yr-1.

L68: Consider whether it is really necessary to abbreviate freshwater as "FW" in the text. The less abbreviations, the easier it will be to read.
Agreed, we removed all abbreviations.

L80: Delete "MAR henceforth", the abbreviation has already been given on the previous line "(MAR)"
Thanks for spotting this, it has been adjusted.

L93-98: If the RACMO simulations exist back to 1940 at a 5.5 km resolution, could you not use the whole time series by conducting the statistical downscaling of Noël et al. (2016)? Or what is the reason for only using 1958-present? Also, why did you choose to not statistically downscale the tundra?
That is a fair comment. Unfortunately, the RACMO run that extends back to 1940 has not yet been statistically downscaled. For consistency, we changed the RACMO tundra data back to be the same run as the RACMO run starting in 1958 (See Figure 2 in this document). This is now explained more clearly in the Methods.
Regarding the downscaling of tundra runoff, we clarified this in the revised text:
Whereas runoff from the GIC and GrIS can be easily statistically downscaled because of the strong elevation dependency of melt and hence runoff, as described by Noël et al. (2019), this is more challenging for the tundra and therefore not done yet. Tundra has much lower runoff values (from seasonal snow melt) than ice-covered regions (i.e. from ice and snow melt), therefore statistical downscaling based on elevation gradients must be applied separately. This has not been done yet, and therefore a 1 km runoff grid was obtained by nearest-neighbour interpolation of the original 5.5 km resolution data.

L98: Is there no dataset from MAR of the tundra runoff which you can compare to? This is the only variable where only one dataset is used

Good suggestion. In the revised MS we now include the annual MAR tundra runoff (See Figure 2: adjusted in this document), and also update this in Figure A3. We add the following text to the discussion:

We find that tundra runoff in RACMO is 36% higher than in MAR (175 vs 128 Gt yr-1, averaged over 1990-2023). Differences are especially large in NO and NE, where the difference is 8 Gt yr-1 (47%) and 13 Gt yr-1 (43%), respectively. Using MAR tundra runoff thus results in a decrease of tundra runoff contribution to the total freshwater input in fjords to 9% and 17%, respectively. This shows that in MAR the contribution of tundra runoff remains significant, but the large model differences require further investigation.

We also explored CARRA tundra runoff, but after consulting with the CARRA support team we decided not to use it for this application because the output fields called runoff did not represent (hydrological) runoff.

Please see comment above on GIC and GrIS runoff (L93-98).

Table 1: Isn't the tundra runoff also interpolated onto a 1 km resolution? There is no arrow in the table like for the runoff.

It is bilinearly interpolated, but not statistically downscaled (please see our reply to the L93-98 comment above). We changed the caption to clarify this as follows:

Statistically downscaled data are indicated with an arrow.

Section 2.2: consider removing the adjustment you added to Slater et al. (2020), as it will be easier to compare to other studies if the same borders are used.

We think there are good reasons for the revisions we applied, please see our extensive answer to "RC 1: L129-131", repeated below:

"We understand your concern regarding the many different basin definitions that are being used in the community. The main reason for doing so is that, for the purpose of this paper, we are 'in-between' applications for land, ocean and ice. This study has the purpose of comparing freshwater sources from the perspective of fjords and climate, as opposed to hydrological basins (on land), ocean basins or glaciological basins. During our research phase, we concluded that using any of the existing basin definitions would not fit our research needs, and therefore a dedicated basin definition was deemed necessary, despite the disadvantage of creating another basin definition as you point out. We followed, as much as possible, existing definitions. We use as a basis the ocean-based basin divides used by Slater et al. (2020), which in turn represents an adjustment of the ice basin divides used by Mouginot et al. (2019). However, the area of CE is very small in Slater et al. (2020), as it was created as a 'transition' sector between colder and warmer ocean waters, while not cutting the Kangerlussuaq glacier in half (Fig. 1). This makes NE six times larger than CE, making the regions not comparable anymore. Furthermore, the Mouginot et al. (2019) - CE/SE boundary over the tundra cuts fjords

in half, which is unsuitable for our purposes. To compensate for both disadvantages, we chose to follow the 'ice divide' boundary from Mouginot et al. (2019) to differentiate between CE/SE. Over the tundra, we manually adjusted the CE/SE boundary to follow the hydrological basin boundaries over land (see Figure 1 in this document, and Zenodo database, see comment "In addition to using standard regions,"). Figure 6 in the MS confirms that our updated basins have the desired inter-distinguishable hydrological and climatological characteristics, indicating that the chosen regions effectively capture variance across Greenland. We will provide information (see below) to increase reproducibility, and to make the outcomes comparable to other studies. We revised the manuscript as follows:

Because this study considers freshwater fluxes into fjords, which differs from most hydrological, glaciological and oceanic applications, Greenland is divided into seven climatologically distinct regions (Fig 1): North (NO), North-East (NE), North-West (NW), Central-East (CE), Central-West (CW), South-West (SW) and South-East (SE). These are based on the seven land/ice basins from Slater et al. (2020) that in turn are based on the Mouginot et al. (2019) ice divides. We made one adjustment by moving the boundary between SE and CE northward, following the Mouginot et al. (2019) ice divide, to follow tundra hydrological catchments, as well as to make the regions more comparable in size."

L150-51: I'd remove this line: "the Total average FW flux since 2010 is 1239 (± 180) Gt yr −1, calculated to compare to other studies in Sect. 4.1". The number can just be provided in Sec 4.1:

Thank you for this suggestion. Although we understand your point, we also believe this total average freshwater flux value is an important result that belongs in the Results section (L150-51), not just in the Discussion. We feel it's appropriate to introduce the core finding here where readers would expect to see it.

L164-65: change to "Basal melt accounts for a maximum of 3± 0.6 % (March)", to clarify that march is the month with the highest contribution from basal melt

Thanks for the good suggestion! We incorporated the statement Basal melt accounts for a maximum of 3 ± 0.6 % (March) into the text

Section 3.2: Information on the trend of the freshwater components are mostly given for NO. Can you write this for the other areas too?

There is one mention of a trend in this section, which we have removed this to make it more consistent with the description of the other areas.

L173-97: Since the discussion is mostly about how much each component contributes to the total runoff for each area, I would suggest changing the numbers in this section to percentages

This is a good suggestion, and the percentages can be found in Table A3.

L186: change "in line with earlier studies" to "in line with an earlier study" (or alternatively, have more than one reference)

We changed it to: in line with earlier work

**L188: delete "now"**
We changed this to:
In recent years,

**L196: what does "(O (10 km))" mean?**
We removed it.

**L232-235: this is a results, I suggest moving to section 3.2**
We moved this to section 3.2:
Annual freshwater fluxes show large variations between the regions, but on average fluxes per sector are 251 Gt yr-1 (SE), 178 Gt yr-1 (SW), 179 Gt yr-1 (CE), 183 Gt yr-1 (CW), and for the northern regions 119 Gt yr-1 (NE), 146 Gt yr-1 (NW), and 88 Gt yr-1 (NO) between 1990 through 2023 (Fig. 4).

**L235-36: "The average annual rate is slightly smaller than the total Arctic FW flux found by Bamber et al. (2018) (1300 Gt yr −1 since 2010), which included non-Greenland ice caps." - I would delete this sentence, since it is not calculated for the same region.**
Indeed, this study does not cover the same region, but for context we still find it relevant to compare, e.g. as an upper bound. We now mention this specifically in the revised text as:
The average annual rate is slightly smaller than the total Arctic freshwater flux found by Bamber et al. (2018) (1300 Gt yr$^{-1}$ since 2010), which included non-Greenland ice caps. The region we study is a subset of the broader Arctic region analyzed by Bamber et al. (2018), making their estimate useful for context and as an upper bound for our values.

**Section 4.1: The results could also be compared to Mankoff et al. (2020)**
We have changed the text to the following:
We find higher values for tundra runoff than Bamber et al. (2018) (150 vs 80 Gt yr-1), and then Mankoff et al. (2020) (100-130 Gtyr-1).

**L236-37: "GrIS runoff between 2010-2016 is higher in Bamber et al. (2018) and Igneczi and Bamber (2024) than estimated in this study." → this sentence is too vague, please provide numbers. And why is it only compared for 2010-16, aren't both studies for longer periods?**
We agree that this was unclear. To clarify this, we changed the text in the revised manuscript to:
Average annual rates are compared for 2010-2016, a period with a relatively smaller trend than preceding years and covered by all studies in this comparison.

**L239: which period is the comparison with bamber et al. (2018) from?**
Please see comment above.

L240-41: ".. similar to values found in this study using RACMO output." - this is too vague, please provide the numbers

The numbers are given in results in section 3.1 L147. We will refer back to this section:
We find higher tundra runoff than Bamber et al. (2018) (150 vs 80 Gt yr-1). Igneczi and Bamber (2024) estimated a total tundra runoff based on MAR between 140-160 Gt yr-1 (1950-2021), similar to values found in this study using RACMO output (Sect. 3.1).

Section 4.2: I am not sure what the added value of the MoA is here. Since you are discussing the fraction of the total runoff, it seems obvious that regions with less melt will have a higher contribution from the solid ice discharge and regions with high melt with have a smaller contribution from the solid ice discharge. And regions with high melt will have a higher MoA, while regions will less melt will have a smaller one. So MoA does not seem to provide any additional information on what is actually happening beyond the obvious – or do I misunderstand what the meaning of the parameter is? Please clarify

Thanks, we will in the revised text better explain what the added value of the MoA analysis is. Please also see our answer to "RC 1: Sect 2.3" or changes to MS repeated below:

"Sect 2.3: The MoA ratio has been used previously in firn studies to determine the climatic conditions under which melt would generate runoff in the accumulation zone. Previous work has identified a theoretical MoA ratio threshold between 0.6 and 0.7, indicating the onset of runoff (Pfeffer et al., 1991; Braithwaite et al., 1994). More recently, MoA has been used to predict when melt ponding starts on Antarctic ice shelves ( van Wessem et al., 2023). In this study, MoA is used as a climatological indicator over the ice sheet, which we hypothesise is highly relevant for the partitioning between solid ice discharge and liquid water runoff into fjords. MoA does not directly depend on runoff, but on melt as well as snowfall; the relative regional sizes of ablation and accumulation areas and the potential for meltwater buffering (through snowfall) also become important.

L255: To our knowledge, no studies have identified a strong link between MoA and freshwater input fractions in Greenland fjords. In contrast, freshwater input fractions into fjords poorly correlate with temperature, melt or snowfall. This novel result will facilitate the interpretation of e.g. future changes in the distribution of freshwater fluxes in terms of climate change."

L251-52 / Figure 6(c-d): I am not sure what the point is of Figure 6c, d. Since runoff and ice discharge are the main contributors, won't these always have a linear relationship? if there is a high contribution from ice discharge (e.g. 70%), of course the contribution from runoff has to be low. And when there is a high contribution from runoff, the contribution of ice discharge cannot be high. I suggest deleting this sentence/these figures, unless there is something important I am missing?

We clarify in the revised text as follows: Furthermore, the regional fractions to total freshwater input of both GrIS and tundra runoff decrease with increasing fraction of solid ice discharge (Figs. 6c,d). Yet, no such relation is found for fraction of GIC runoff or precipitation. There are more sources than solid ice discharge and meltwater runoff alone, such as ice cap runoff,

tundra runoff and precipitation. This means that the relationship between two sources is not a priori linear, and we find that there is such a relation between all pairs of source fractions.

L262: add the relative distribution in Bamber et al. (2018)
We have change the text as follows:
Annual mean tundra runoff is on average 15% of the total runoff (1990-2023), exceeding the relative contributions estimated in previous work (9-11% estimated from Fig. 3 in Bamber et al., 2018). Seasonal contributions from tundra runoff and fjord precipitation to Greenland-wide freshwater fluxes can reach up to 35% and 11%, respectively,

L264: what is a "relatively high contribution"? Can you provide percentages?
We have changed the text to include percentages:
The south and east of Greenland have relatively high precipitation rates (Fettweis et al., 2020; van Dalum et al., 2024), which leads to the regions CE, SE, SW, and NE having a relatively high contribution of fjord precipitation to the total freshwater input (monthly percentages up to 22, 12, 20, 22%, respectively), especially from October until April, when runoff is small (Fig. 5).

L337-341: please cite the datasets in addition to the papers
We have now included citations for the data sets with a DOI in the paper.

Figure 3: why does (a) and (b) show 1990-2004 and 2005-2023 instead of the whole period? This is not discussed in the text. I suggest either only showing the values for the whole time period (1990-2023), or alternatively add some text about the difference between the two time periods.
Thanks for pointing this out, we have changed the text:
There is an increase in summer GrIS runoff in the latter half of the period (2005-2023) compared to the first half (1990-2004), along with a rise in monthly solid ice discharge (Fig. 3a).

Figure 6: I think the "R" on the figures should be "r", like in the caption?
Yes thanks for pointing it out, we changed this.

[Figure]

Figure 1: Masks of different surface types at 1x1 km2 resolution representing: the contiguous ice sheet (GrIS, orange), tundra (green), glaciers and ice caps (GIC, violet) and fjords (red). Solid black lines delineate the seven climatological regions used in this study, dashed olive green lines delineate the Slater et al. (2020) basins and dashed cyan blue lines delineate the Mouginot et al. (2019) basins (see legend in the upper left corner).

[revised manuscript text omitted]

---

## Author Response (AR2)

**Vries et al. (2025) - Seasonal … - Minor revisions**

June 12 2025

First of all, we would like to thank Ken Mankoff for his time in reviewing the revised manuscript and pointing out some remaining concerns. Responses to the issues raised (in blue) are written in black, and changes in the manuscript are written in red.

The authors have addressed my concern over use of MOA.
They have explained why they are using yet-another-region definition. I don't think it's a good idea or particularly helpful or useful to the study or the community, but it's not a flaw in the study. Most of the minor issues I raised have been addressed. I have a few remaining concerns.

** Flux vs mass flow rate

I strongly urge the editor to make the correct use of technical terminology a requirement for publication. Text and figure captions.
I would be very happy to be proven wrong on this opinion. I don't want to be that annoying reviewer who insists on this change, nor the editor who is continuously asking for this change. It's tiring. But I've looked into this from Wikipedia to ISO standards to textbooks, and I have not found any evidence to support use of the word 'flux' with dimensions 'mass / time' (or 'volume / time'). Flux dimensions are 'mass / time / length^2'. Without area, the correct term is 'mass flow rate' (or 'volume flow rate'). Please let me know if you have solid evidence to the contrary (i.e., not just other earth science papers which may be perpetuating an error).
Based on my research, the use of flux is incorrect when attached to units Gt yr^{-1}. Citing other papers that used it incorrectly isn't a good idea. Most of our community uses this term incorrectly, which may be why the authors want to continue this tradition. It's a bit uncomfortable to start using a term that most people are not using. But it's the right thing to do. Words have meanings. I'm OK with language evolving over time, but technical terminology has a higher bar than colloquial terms.
Throughout the paper the authors now make frequent use of "input" instead of flux. That's an improvement, but 'flux' could (should not but could) actually be used in some places throughout the text where you're talking about broader concepts and there are no associated units. 'input' is better. 'mass flow rate' is the correct term because the units are implied from elsewhere in the document. But 'flux' would not be terrible. The places where it now remains in the paper are the worst places for it to remain - immediately next to units. One of the two should change. I suggest changing the word, rather than re-doing all the analysis to change the units, because the correct units (Gt yr^{-1} km^{-2}) are not useful.

Following the reviewer's recommendation, we have changed the figure labeling to replace 'Flux' with 'Freshwater input' on the y-axes for improved clarity and consistency in the terminology, as well as where the word flux is next to a unit.

** All sources

I wrote:

"L164: Basal amount is mostly steady state. What is the goal of reporting this small amount on this month? What is the significance of this sentence?"

Reply was:

"It is true basal melt is small with low temporal variability. The sentence is included because all terms are discussed, and we want to be consistent by discussing every term."

I disagree that all terms are discussed. Please see graphics at https://github.com/mankoff/sankey I believe this work is missing at least frontal retreat, grounding line retreat, and sub-shelf melt if not other terms. These combined are ~80 Gt/yr in Greenland, or ~3x grounded ice basal melt.

I recommend removing basal melt from the table, adding a paragraph discussing other sources and/or limitations where you list basal melt and other terms. This could be done where you discuss the temporal issues introduced by sea ice.

We thank the reviewer for highlighting this. Basal melt is included in our study because it constitutes a component of freshwater input for which estimates are available, hence providing a more complete picture compared to leaving it out. To provide improved context, we now extend the discussion in the revised manuscript from line 352 onwards, as follows:

Solid ice discharge is determined by applying fixed flux gates, which comes with the advantage that the solid ice discharge in this study is consistent with earlier work (Mouginot et al., 2019; King et al., 2020; Mankoff et al., 2020). However, the disadvantage is that we neglect solid ice discharge due to systematic glacier front retreat, which is estimated to average 42 Gt yr-1 since 2000 for the total ice sheet (Greene et al., 2024), i.e.  similar to or larger in magnitude than basal melt over the studied period. Because the glacier front advances in winter and retreats in summer (63±6 Gt seasonally (Mankoff et al., 2020, Greene et al., 2024)) — the use of fixed flux gates likely leads to an underestimation of solid ice discharge in summer and an overestimation in winter. This error is however less important for a study addressing freshwater input assuming fixed outlines of the fjord, like we do here, than one addressing the freshwater budget of fjords.

** Figure colors

Please add

import matplotlib.pyplot as plt

plt.style.use('tableau-colorblind10')

Or something similar and remake your figures so that they can be viewed by people who may not be able to distinguish current colors.

We share the reviewer's concern for accessibility of the manuscript. For this reason, we have ensured that the figures are well interpretable by checking our figures using the color blindness simulator Coblis  (https://www.color-blindness.com/coblis-color-blindness-simulator/) for Anomalous Trichromacy and Dichromatic view.

---

## Author Response (AR3)

Response to the editor for manuscript "Seasonal … fjords" by Vries et al. (2025)
2025-06-24

We thank the editor for their consideration for colorblind-friendly colormaps. We have changed the colors in all figures to "seaborn colorblind palette"  and added markers where we deemed necessary. Please find a screenshot of the green and red blind deficiency simulator below for Figure 3 below.

Drag and drop or paste your file in the area below or: [ Browse... ] No file selected.

*Trichromatic view:* *Anomalous Trichromacy:* *Dichromatic view:* *Monochromatic view:*

○ Normal    ○ Red-Weak/Protanomaly    ○ Red-Blind/Protanopia    ○ Monochromacy/Achroma

○ Green-Weak/Deuteranomaly    ● Green-Blind/Deuteranopia    ○ Blue Cone Monochromad

○ Blue-Weak/Tritanomaly    ○ Blue-Blind/Tritanopia

Use lens to compare with normal view:  ● No Lens   ○ Normal Lens   ○ Inverse Lens

Reset View   Open simulated image in new window

[Figure]

Drag and drop or paste your file in the area below or: [Browse...] No file selected.

*Trichromatic view:*  *Anomalous Trichromacy:*        *Dichromatic view:*            *Monochromatic view:*

○ Normal        ○ Red-Weak/Protanomaly      ◉ Red-Blind/Protanopia       ○ Monochromacy/Ac
                ○ Green-Weak/Deuteranomaly   ○ Green-Blind/Deuteranopia    ○ Blue Cone Monoch
                ○ Blue-Weak/Tritanomaly      ○ Blue-Blind/Tritanopia

Use lens to compare with normal view: ◉ No Lens  ○ Normal Lens  ○ Inverse Lens
Reset View   Open simulated image in new window